# Validity and Consistency of the Arabic Version of the Eating Disorder Examination Questionnaire (EDE-Q) among Saudi Adults

**DOI:** 10.3390/healthcare11071052

**Published:** 2023-04-06

**Authors:** Khalid Aldubayan, Khloud Ghafouri, Hiba Mutwalli, Hebah A. Kutbi, Walaa A. Mumena

**Affiliations:** 1Department of Community Health Sciences, Collage of Applied Medical Sciences, King Saud University, P.O. Box 10219, Riyadh 11433, Saudi Arabia; 2Departmrnt of Clinical Nutrition, Collage of Applied Medical Sciences, Umm Al-Qura University, P.O. Box 715, Makkah 21955, Saudi Arabia; kjghafouri@uqu.edu.sa; 3Department of Clinical Nutrition, College of Applied Medical Sciences, Imam Abdulrahman Bin Faisal University, P.O. Box 1982, Dammam 31441, Saudi Arabia; hsmutwalli@iau.edu.sa; 4Clinical Nutrition Department, Faculty of Applied Medical Sciences, King Abdulaziz University, P.O. Box 80215, Jeddah 21589, Saudi Arabia; hkutbi@kau.edu.sa; 5Clinical Nutrition Department, College of Applied Medical Sciences, Taibah University, P.O. Box 344, Madinah 42353, Saudi Arabia; wmumena@taibahu.edu.sa

**Keywords:** Eating Disorders, Eating Disorder Examination Questionnaire, Validation, Consistency, Confirmatory Factor Analysis, Exploratory Factor Analysis, Saudi Arabia, Arabic version

## Abstract

The prevalence of eating disorders (EDs) is growing, and early screening is important to prevent related health complications. The Eating Disorder Examination Questionnaire (EDE-Q) has been widely used as a diagnostic tool to identify cases of EDs; however, a validated Arabic version of the tool is needed to help in the screening process of EDs. The aim of this study was to validate the Arabic version of EDE-Q. A cross-sectional study included a sample of 549 adults, who were recruited mainly from the four major provinces in Saudi Arabia. A forward–backward translation method was conducted, and then the tool was validated using the confirmatory factor analysis (CFA). The dataset was split for further convergent analysis using exploratory factor analysis (EFA) and CFA. The results of CFA from the main dataset did not support the four-factor original EDE-Q. The results of EFA from the first data-split suggested a three-factor EDE-Q-14 Arabic version. This was supported by the results of CFA of the second data-split. A total of five items were allocated in each *shape and weight concern*,* and restraint* component, with correlations ranging from 0.969 and 0.462 and from 0.847 to 0.437, respectively. A total of four items were allocated in *eating concern,* with correlations ranging from 0.748 to 0.556. The internal consistency of the global and the three subscales were high, with Cronbach’s α ranging from 0.762 to 0.900. Findings of the current study suggest that the Arabic version of the EDE-Q-14 is a valid and reliable tool to screen for EDs among adults in Saudi Arabia.

## 1. Introduction

Eating Disorders (EDs) are mental health issues that can affect eating behaviors and body weight [1]. Forms of EDs are described in the International Classification of Diseases and Related Health Problems (ICD) and the Diagnostic and Statistical Manual of Mental Disorders (DSM). The most commonly used classification is the fifth version of the DSM, DSM-5, which characterizes three typical ED forms: anorexia nervosa, bulimia nervosa, and binge eating disorder. Other forms of EDs are referred to as other specified feeding or eating disorders, which comprise all other atypical forms of EDs [2].

EDs have a significant impact on health and contributes to high rates of mortality [3] and health-related complications, including suicidal behaviors [4], anxiety [5], low blood pressure [6,7], and severe dehydration in anorexic and patients [8]. EDs have been also linked to obesity [9], electrolyte imbalance, and type 2 diabetes mellitus in patients with bulimia nervosa and binge eating disorder [10,11]. Therefore, early screening for EDs is considered vital to prevent health-related complications. A number of diagnostic tools have been used in community and clinical settings, including the Eating Attitude Test (EAT-26) [12], Sick, Control, One, stone, Fat, Food (SCOFF) questionnaire [13], eating disorder screen for primary care (ESP) [14], and Eating Disorder Examination (EDE) tools [15]. For non-clinical and clinical screening for EDs, EDE has been considered the “gold standard” tool to identify patients with EDs [16,17]. The Eating Disorder Examination Questionnaire (EDE-Q) [18], an updated version of the EDE, was issued in 2014 [19]. It is a self-reported questionnaire composed of 28 items, which has the ability to measure underlying aspects of ED psychopathology in young teenagers [20,21]. Both EDE and EDE-Q tools were validated and used among some populations, and results suggested their validity and reliability for screening patients for EDs [22].

Recent reports documented a steadily increasing prevalence of EDs in many countries [23,24]. Therefore, many studies aimed to assess the validity of a variety of ED tools in different languages for early screening and management, including Malaysian [25], Persian [26], Italian [27], and Spanish [28]. In the Middle East, the Arabic version of the SCOFF questionnaire has been validated using data collected between 2008 and 2009 [29]. However, the SCOFF instrument has been limited by its low sensitivity among the young population, wherein a high proportion of individuals with EDs may not be identified by the SCOFF [30]. On the other hand, the EDE-Q has been previously shown to be a valid and reliable tool to screen patients for EDs [22].

Studies have shown that EDs may particularly occur in cultures experiencing rapid socioeconomic and cultural transitions [31,32], which could be expressed by cultural adaptation of lifestyle, language, and beliefs of other countries [33]. Meanwhile, communities of the Middle East have been recently experiencing a greater contact with populations of other countries [34,35] and exposure to Western media [36], which led to increased popularity of dieting and EDs [37]. In fact, a recent study estimated the prevalence of individuals at high risk for EDs in Middle Eastern countries, including Saudi Arabia, as ranging between 2 and 54.4%, wherein increased obesity, media use, western influences, and affluence were found to be significant correlates to EDs [38].

In Saudi Arabia, a number of research studies have been conducted to assess the prevalence of EDs and its related factors among young individuals using different ED screening tools [39,40,41,42,43,44,45]. However, the majority of the studies aimed to assess the prevalence of EDs without examining the validity of the tool among the Saudi population. This is further supported by a study conducted in 2020 that aimed to investigate the prevalence of individuals at high risk for EDs among the Arab communities, including Saudi Arabia, using the data of 81 studies. The study highlighted methodological shortcomings due to the absence of a culturally sensitive, validated tools for the Arab communities [38]. Therefore, the aim of the present study was to validate the Arabic version of the EDE-Q among adults in Saudi Arabia.

## 2. Materials and Methods

This cross-sectional study aimed to validate the Arabic EDE-Q among Saudi adults residing in the most populated provinces in Saudi Arabia (Riyadh, Makkah, Madinah, and AlSharqiya); participants living in all other 13 provinces were also invited. The EDE-Q was designed to be self-reported, and it was validated from the interview-based EDE (23). The EDE-Q consist of two types of data: frequency data as in term of number of episodes (6 items; from 13 to 18), and subscale scores as in term of severity of eating disorder (22 items in 4 subscales: restraint, eating concern, shape concern, and weight concern) (28). The adapted EDE-Q has gone through forward–backward translation process. Two native Arabic speakers, one expert in the field of nutritional assessment who is fluent in English and one with expertise in English translation, conducted the forth translation of EDE-Q from English to Arabic. An independent bilingual specialist in both Arabic and English has carried out the backward translation of the questionnaire. Another two expert researchers, who did not participate in the forward translation, have critically reviewed the backwards-translated questionnaire against the original version. The wording of the items was approved after minor grammatical changes made to the translated Arabic version.

The translated Arabic version of the EDE-Q was pilot tested among 10 participants with equal proportions of males and females to ensure clarity of the items. The ages of the participants ranged from 18 to 60, with varied educational level ranging from high school to Ph.D level. For each item, a scale from 0 (not clear) to 1 (clear) with a comment box was introduced for the participants. There were a few comments suggesting minor changes in words and terms, and these were addressed accordingly, preserving the same meaning.

The study was conducted between January 2021 and August 2022. Arabic-speaking adults residing in the aforementioned four major provinces were targeted in our study. The final version of the EDE-Q was distributed online through major universities’ email portal and varied social media platforms. Participants who were adolescents or lived outside Saudi Arabia were excluded from this study. According to Monte Carlo sample size calculation method, a sample size of 200 has a high percentage of convergence for 10 variables and 3 factors [28]. According to the final EDE-Q’s items, a sample size of 560 participants was needed for the present study.

The data was analyzed by using the Statistical Package for Social Sciences (SPSS) software version 23 and Amos structural equation modeling (SEM) version 26. Demographic data were illustrated in counts and percentages. The normality of the data was investigated by Kolmogrov–Smirnov test. The homogeneity of variance was tested by Levene’s test. Confirmatory factor analysis (CFA) with the maximum likelihood estimates was performed to test the original four-factor model of the EDE-Q for the main dataset and in each of the samples of four major provinces (Riyadh, Makkah, Madinah, and AlSharqiya). The comparative fit index (CFI) and Tucker–Lewis index (TLI; high fit > 0.95, acceptable fit = 0.90–0.95, poor fit < 0.90) were indices in addition to the root mean square error of approximation (RMSEA; high fit < 0.06, acceptable fit = 0.06–0.08, poor fit > 0.08) [46,47]. Data were split into two independent datasets with equal proportions of participants provinces for further analysis using exploratory factor analysis (EFA) and CFA. EFA with extraction method principal axis factoring (PAF) with Oblimin rotation method was utilized for the first half of dataset after split to examine the closeness of items and factors of EDE-Q among the participants. Sampling adequacy for EFA was evaluated by Kaiser–Meyer–Oklin (KMO) test. Cronbach’s alpha test was used to assess the internal reliability/consistency of the EDE-Q’s items and subscales. CFA was performed for the second half of dataset after split to confirm the suggestion SEM result of EFA. The significance level was determined at alpha 0.05.

## 3. Results

The final Arabic version of the EDE-Q has reached to 626 participants. After data cleaning, 549 participants completed the questionnaire with an 88% response rate. All completed questionnaires were included in the analysis. The majority of participants were Saudis (95.4%), females (83.6%), and single (70.3%), while over two-thirds of the participants were holding a bachelor’s degree (67.9%), Table 1. All participants were almost equally distributed from the main provinces in Saudi Arabia (Riyadh, Makkah, Madinah, and AlSharqiya). Nearly half of the participants were not specialized in health, and 45.0% had a BMI that fell within the normal range weight status. The homogeneity of the two halves of the dataset after the split was tested.

After 100,000 iterations with a minimum of 15 achieved, CFA for Saudi adults as whole and in each four major provinces failed to converge to the original four-factor model of EDE-Q. In all samples, CFI/TLI < 0.90 and RMSEA > 0.10 suggested poor fit, Table 2. The four-factor model for the Saudi adults as a whole and their standardized estimates are illustrated in Figure 1.

Fourteen out of twenty-two items of the EDE-Q were allocated into three subscales by performing exploratory factor analysis (EFA) on first half split of dataset (n = 275) with the extraction method Principal Axis Factoring (PCA) and the rotation method Oblimin with Kaiser–Meyer–Olkin (KMO). A total of 5 five were allocated in shape and weight concern components with correlations ranging from 0.969 to 0.462 and Eigenvalues equal to 6.211. The other items were allocated in restraint (five items) and eating concern (four items), with correlations ranging from 0.847 to 0.437 and from 0.748 to 0.556, respectively, and Eigenvalues equal to 1.963 and 1.078, respectively. The final percentage of variance was explained by approximately 66%. The results of EFA are reported in Table 3. The sampling adequacy by Kaiser–Meyer–Oklin (KMO) was 0.907, and Bartlett’s test of sphericity was 4083.5, *p* < 0.001. The correlations among the three-factor model are reported in Table 4. According to the Pearson correlation test, the body mass index (BMI) of participants was positively moderately correlated with the five items of shape and weight concern components, ranging from 0.347 to 0.553, with *p*-value > 0.001. In addition, positive slight to moderate correlations were found between BMI and both restraint and eating concerns, ranging from 0.158 to 0.341, with *p*-value > 0.01. According to cross-tabulation results, no associations were found among participants socio-demographics and their scores of EDE-Q14 Arabic version except their speciality, Table 5.

The Cronbach alpha values of the EDE-Q Arabic version and the three subscales were 0.900 for global, 0.891 for shape and weight concern, 0.839 for restraint, and 0.762 for eating concern. The distribution of the EDE-Q-14 Arabic version was normally distributed with slight positive skewness (Mean = 0.62) and slightly heavily tailed negative kurtosis (Mean = −0.82). Similar results were found in all subscales except Weight and Shape Concern and Eating Concern. The distribution of Weight and Shape concern was moderately light-tailed negative kurtosis (Mean = −1.51), and Eating Concern was slightly positively skewed (Mean = 1.08) with slightly heavily tailed positive kurtosis (Mean = 0.02). The results of consistency and distribution are presented in Table 6.

After 100,000 iterations with a minimum of 8 achieved, the CFA for Saudi adults of the second half of the split dataset (n = 274) succeeded to converge to the three-factor model of EDE-Q-14 Arabic version suggested by EFA of the first half of split dataset (n = 275). In samples, CFI/TLI within 0.90–0.95 and RMSEA within 0.06–0.08 suggested acceptable fit, Table 7.

## 4. Discussion

Findings of this study showed a poor fit for a four-factor model (Restraint, Eating Concern, Shape Concern, Weight Concern) according to the CFA results. The dataset was split into two halves for further convergent validity test (EFA for the first half split n = 275 and CFA for the second half split n = 274). The results of EFA indicated wide correlation values ranged between 0.969 and 0.462 for the five items that were allocated in shape and weight concern component. Seven questions that had correlations less than 0.4 were removed from the three-factor model to reach a good fit for the Saudi adults. Factor loading > 0.4 was determined as stable and acceptable for the model in eating disorders [48,49,50]. These seven questions were as follows: Q6 flat stomach, Q8 preoccupation with shape or weight, Q10 fear of weight gain, Q12 desire to lose weight, Q22 importance of weight, and Q23 importance of shape. Narrower correlation values ranging between 0.847 and 0.437 were found in restraint with five items allocated, while and values between 0.748 and 0.556 were found in eating concern with the four items allocated. A question in eating concern component was removed due to correlation value < 0.4. This question was Q19 eating in secret. No cross-loadings were found in the three-factor model to determine the number of factors to retain. The internal consistency of the global and three subscales were high, with Cronbach’s α ranging from 0.762 to 0.900.

Exploratory factor analysis (EFA) suggested a good fit for a 3-factor model (Shape and Weight Concern, Restraint, Eating Concern) and 14-item with adequate sampling (KMO = 0.907, *p* < 0.001). Our findings of the EDE-Q Arabic version did not support the use of the original EDE-Q four-factor model (Restraint, Eating Concern, Shape Concern, and Weight Concern) [19]. In the EDE-Q Arabic version, the two subscales (Shape Concern and Weight Concern) were merged into one subscale with five items compared to the original EDE-Q. The results of EFA were supported by CFA. Confirmatory factor analysis (CFA) revealed a good fit of the 3-factor model, with 14 items suggested by the EFA. In fact, a similar approach has been also conducted for the EDE-Q after translation and validation among nonclinical general population such as the Malaysian version of the EDE-Q [25], the Sweden version [51], and the South Korean version [52]. All of these versions had a similar good fit for a three-factor model. Unlike our study, these versions were validated among narrower target populations such as adolescents or university students. On the other hand, versions such as the Spanish [28], Turkish [53], Italian [27], and Fijian [54] were similar to the original EDE-Q as a four-factor model. Their population were nonclinical and varied from a wide age range including adults to a narrow age range focusing on teenagers. In addition, two studies with similar models were validated among clinical and special populations such as anorexic and bulimic patients and athletics [55,56,57]. One large-scale Mexican study has looked over general population in different regions as in the present study [57].

Furthermore, our 3-factor model has 14 items compared to 22 items of the original EDE-Q, as 8 items were removed according to EFA and CFA results of 2 halves split dataset. Similar removing were found in versions including the Italian [27], Sweden [51], and Turkish [53]. Both Sweden and Turkish versions had 13 items in their model compared to the original EDE-Q-22, whereas the Italian version had only 7 items in its model. In addition, our study suggested that BMI has a positive slight to moderate association with the components of the EDE-Q-14 Arabic version. This association was also found in the Sweden version [51]. Our results also suggested a possible association between participants specialty and their scores of EDE-Q-14 Arabic version, as participants with health specialty tend to have lower risk of EDs compared to non-health specialty. A Saudi study found that a low prevalence of EDs among health specialty college students [58].

The results of our study suggest high internal consistency to the EDE-Q-14 Arabic version and all three subscales. Similarly, studies with goodness of fit to a three-factor model had adequate Cronbach’s α values for their global and three subscales [51,52,59], whereas studies with goodness of fit to a four-factor model similar to the original EDE-Q-22 had Cronbach’s α values ranging from adequate to high for their global and four subscales [27,54,59,60].

The strengths of this study include the adequate sample size collected mainly from four heavily populated provinces, and the wide range of age groups recruited. In addition, this study was the first to translate and validate the original EDE-Q into Arabic using both CFA and EFA. However, some limitations should be addressed. First, this study had a high proportion of females compared to males. However, females in the Arab world have been frequently documented to be at higher risk for EDs [35,61,62] because of many reasons, such as higher vulnerability to body dissatisfaction [61,63,64] and disturbed eating behaviors [61] compared to males. Second, this study was a population-based one, and a validation at the clinical setting was not conducted. Third, data concerning weight and height were self-reported, which may create biases such as over- or underreporting.

## 5. Conclusions

The findings of this study suggest the validity and reliability of the Arabic version of EDE-Q among adults residing in Saudi Arabia. However, discriminant validity of the EDE-Q Arabic version among patients with EDs should be investigated. In addition, future research should investigate the validity and reliability of the Arabic version of the EDE-Q among younger populations. Further studies are required for examining possible risk factors of EDs among Saudi population.

## Figures and Tables

**Figure 1 healthcare-11-01052-f001:**
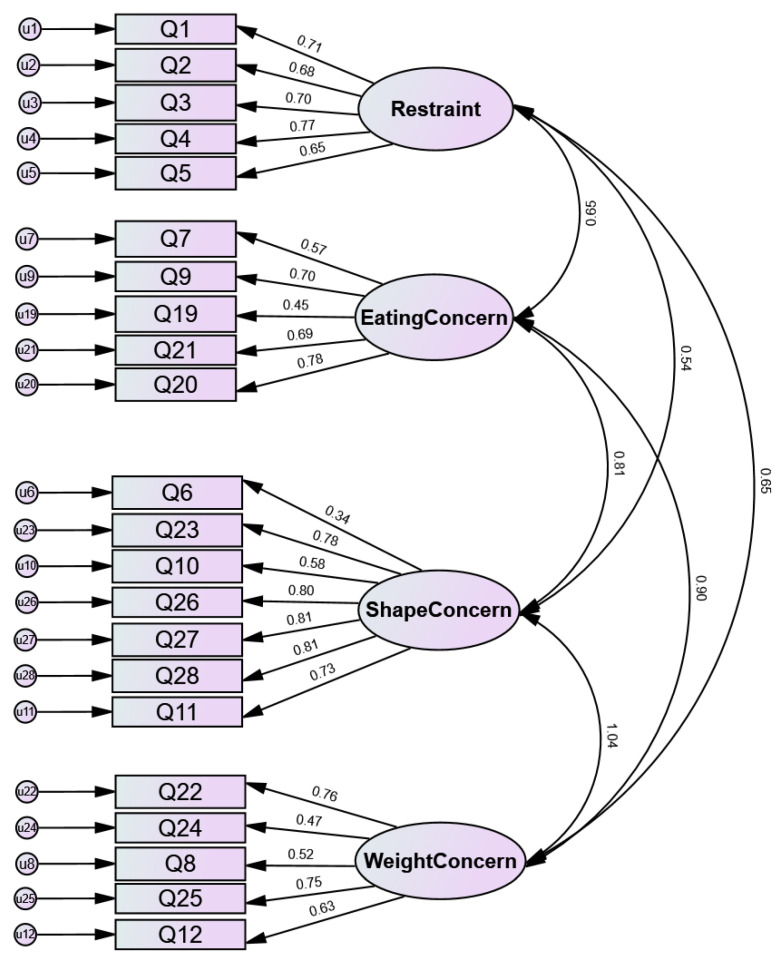
Four-factor model of the original EDE-Q using CFA (Standardized Estimates).

**Table 1 healthcare-11-01052-t001:** Demographics characteristics of the participants.

Variables		Count (%)			
		Main Dataset (n = 549)	First-Half Split for EFA (n = 275)	Second-Half Split for CFA (n = 274)	*p*-Value ^1^
Age	18–19 Years	81 (14.8%)	39 (14.2%)	42 (15.3%)	0.347
	20–24 Years	219 (39.9%)	102 (37.1%)	117 (42.7%)	
	25–29 Years	99 (18.0%)	50 (18.2%)	49 (17.9%)	
	30–34 Years	150 (27.3%)	84 (30.5%)	66 (24.1%)	
Gender	Male	90 (16.4%)	46 (16.7%)	44 (16.1%)	0.832
	Female	459 (83.6%)	229 (83.3%)	230 (83.9%)	
Education	High School	69 (12.6%)	32 (11.6%)	37 (13.5%)	0.917
	Diploma	30 (5.5%)	16 (5.8%)	14 (5.1%)	
	Bachelor	373 (67.9%)	190 (69.1%)	182 (66.4%)	
	Master	59 (10.7%)	29 (10.5%)	31 (11.3%)	
	Doctorate	18 (3.3%)	8 (2.9%)	10 (3.6%)	
Nationality	Saudi	524 (95.4%)	260 (94.5%)	264 (96.4%)	0.310
	Non-Saudi	25 (4.6%)	15 (5.5%)	10 (3.6%)	
Provence	Riyadh	120 (21.9%)	60 (21.8%)	60 (21.9%)	0.997
	Makkah	137 (25.0%)	69 (25.1%)	68 (24.7%)	
	AlSharqiya (Eastren)	132 (24.0%)	66 (24.0%)	66 (24.1%)	
	Madinah	124 (22.6%)	62 (22.5%)	62 (22.6%)	
	Others ^2^	36 (6.5%)	18 (6.6%)	18 (6.7%)	
Health Specialty	Yes	252 (45.9%)	114 (41.5%)	138 (50.4%)	0.106
	No	297 (54.1%)	161 (58.5%)	136 (49.6%)	
Social Status	Single	386 (70.3%)	183 (66.5%)	203 (74.1%)	0.180
	Married	163 (29.7%)	92 (33.5%)	71 (25.9%)	
Weight Status	Underweight (<18.5 kg/m^2^)	74 (13.5%)	42 (15.3%)	31 (11.3%)	0.579
	Normal Weight (18.50–24.99 kg/m^2^)	247 (45.0%)	116 (42.2%)	131 (47.8%)	
	Overweight (25.00–29.99 kg/m^2^)	138 (25.1%)	71 (25.8%)	67 (24.5%)	
	Obese Stage 1 and 2 (30.00–39.99 kg/m^2^)	83 (15.1%)	43 (15.6%)	41 (15.0%)	
	Morbidly Obese (>40 kg/m^2^)	7 (1.3%)	3 (1.1%)	4 (1.5%)	

^1^ *p*-value from Chi-Square test. ^2^ All other 13 provinces: Qasim, Tabuk, Asir, Jawf, Jazan, Hail, Najran, Baha, and Northern Borders.

**Table 2 healthcare-11-01052-t002:** Goodness of fit for the original four-factor model of the EDE-Q tested by CFA of main dataset and four major provinces of Saudi Arabia.

Indices ^1^	Main Dataset (n = 549)	Riyadh (n = 120)	Makkah (n = 137)	Madinah (n = 124)	AlSharqiya (n = 132)
CFI	0.732	0.727	0.635	0.762	0.734
TLI	0.694	0.689	0.584	0.730	0.697
RMSEA	0.133	0.140	0.151	0.139	0.128

^1^ Comparative fit index and Tucker–Lewis index (high fit > 0.95, acceptable fit = 0.90–0.95, poor fit < 0.90); root mean square error of approximation (high fit < 0.06, acceptable fit = 0.06–0.08, poor fit > 0.08).

**Table 3 healthcare-11-01052-t003:** Exploratory factor analysis (EFA) for 14 items from the Eating Disorder Examination Questionnaire (EDE-Q 6.0) (first half split of dataset, n = 275).

Items	Communalities		Factor			Component
	Initial	Extraction	1	2	3	
26 Dissatisfaction with shape	0.747	0.832	0.969			Shape and Weight Concern
27 Discomfort seeing body	0.708	0.749	0.853			
25 Dissatisfaction with weight	0.677	0.702	0.836			
28 Avoidance of exposure	0.635	0.663	0.707			
11 Feelings fatness	0.549	0.561	0.462			
8 Preoccupation with shape or weight	0.372	0.373				
23 Importance of shape	0.332	0.286				
22 Importance of weight	0.331	0.264				
12 Desire to lose weight	0.319	0.326				
10 Fear of weight gain	0.291	0.278				
6 Flat stomach	0.242	0.304				
24 Reaction to prescribed weighing	0.202	0.281				
4 Dietary rules	0.523	0.646		0.847		Restraint
3 Food avoidance	0.455	0.491		0.698		
1 Restraint overeating	0.444	0.502		0.682		
2 Avoidance of eating	0.432	0.455		0.651		
5 Empty stomach	0.443	0.452		0.437		
21 Social eating	0.412	0.528			0.748	Eating Concern
7 Preoccupation with food, eating or calories	0.545	0.548			0.600	
9 Fear of losing control overeating	0.427	0.471			0.585	
20 Guilt about eating	0.527	0.567			0.556	
19 Eating in secret	0.236	0.192				
Eigenvalues			6.211	1.963	1.078	
Percent explained variance			44.362	14.023	7.703	

Notes. Extraction method: Principal Axis Factoring (PAF); Rotation method: Oblimin with Kaiser–Meyer–Olkin KMO = 0.907, Bartlett’s test of sphericity chi-square = 4083.5, *p* = 0.000; factor loading < 0.40 are suppressed.

**Table 4 healthcare-11-01052-t004:** Factor correlation matrix by EFA (first half split of dataset, n = 275).

Factor	Shape and Weight Concern	Restraint	Eating Concern
Shape and Weight Concern	1		
Restraint	0.384 ***	1	
Eating Concern	0.579 ***	0.587 ***	1

*** *p*-value < 0.001 (2-tailed).

**Table 5 healthcare-11-01052-t005:** Cross-tabulation results of the association between participants specialty and their scores of EDE-Q14 Arabic version (first half split of dataset, n = 275).

EDE-Q14			Specialty		Chi-Square	*p*-Value
Components	Questions		Health	Non-Health		
Shape and Weight Concern	Q26 Dissatisfaction with shape	≤15 Days	69	89	0.752	0.386
		>15 Days	54	72		
	Q27 Discomfort seeing body	≤15 Days	72	91	1.958	0.162
		>15 Days	40	70		
	Q25 Dissatisfaction with weight	≤15 Days	71	87	1.855	0.173
		>15 Days	43	74		
	Q28 Avoidance of exposure	≤15 Days	74	95	0.983	0.322
		>15 Days	40	66		
	Q11 Feelings fatness	≤15 Days	74	67	14.500	0.000 ***
		>15 Days	40	94		
Restraint	Q4 Dietary rules	≤15 Days	93	107	7.692	0.006 **
		>15 Days	21	54		
	Q3 Food avoidance	≤15 Days	90	102	7.701	0.006 **
		>15 Days	24	59		
	Q1 Restraint overeating	≤15 Days	79	72	16.284	0.000 ***
		>15 Days	35	89		
	Q2 Avoidance of eating	≤15 Days	94	124	1.201	0.273
		>15 Days	20	37		
	Q5 Empty stomach	≤15 Days	104	122	10.882	0.001 **
		>15 Days	10	39		
Eating Concern	Q21 Social eating	≤15 Days	102	139	0.607	0.436
		>15 Days	12	22		
	Q7 Preoccupation with food, eating or calories	≤15 Days	102	136	1.434	0.231
		>15 Days	12	25		
	Q9 Fear of losing control overeating	≤15 Days	90	83	21.466	0.000 ***
		>15 Days	24	78		
	Q20 Guilt about eating	≤15 Days	92	102	9.666	0.002 **
		>15 Days	22	59		

** *p*-value < 0.01 (2-tailed). *** *p*-value < 0.001 (2-tailed).

**Table 6 healthcare-11-01052-t006:** Consistency and distribution of the EDE-Q Arabic version and items in each component.

Component	No. of Items	Mean (SD ^1^)	Skewness	Kurtosis	Cronbach’s α
EDE-Q Arabic version (Global)	14	2.27 (2.22)	0.62	−0.82	0.900
Shape and Weight Concern	5	2.96 (2.28)	0.09	−1.51	0.891
Restraint	5	2.04 (2.26)	0.77	−0.79	0.839
Eating Concern	4	1.71 (2.10)	1.08	0.02	0.762

^1^ Standard Deviation.

**Table 7 healthcare-11-01052-t007:** CFA results (second half split of dataset)—goodness of fit for the three-factor model of the EDE-Q suggested by EFA (first half split of dataset).

Indices ^1^	Second Half Split of Dataset (n = 274)
CFI	0.938
TLI	0.924
RMSEA	0.079

^1^ Comparative fit index and Tucker–Lewis index (acceptable fit = 0.90–0.95, poor fit < 0.90); root mean square error of approximation (acceptable fit = 0.06–0.08, poor fit > 0.08).

## Data Availability

The raw data supporting the conclusions of this article will be made available by the authors, without undue reservation.

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
