# Peer review of "Validity and Consistency of the Arabic Version of the Eating Disorder Examination Questionnaire (EDE-Q) among Saudi Adults"

_healthcare, 2023, doi:10.3390/healthcare11071052_

Round 1
Reviewer 1 Report
This study aims to validate the Arabic version of EDE-Q. The paper is well-written and easy to read, but there are were several minor grammatical errors which should be fixed. Also, an effort should be made not to avoid repeating the same words to make the narrative more enjoyable to read. I noticed the word “additionally” throughout the manuscript.
The introduction establishes well the motivation for this study.
The sample size calculation is scant on detail. It needs to be embellished so that the reader has a better understanding of by 560 is needed in your study
There is very little information provided about how the sample was obtained, apart from the fact that an online survey was offered to participants in 5 cities. Were there any inclusion or exclusion criteria? Validity is the greatest threat to online studies and the fact that an overwhelming 80% of your sample was female strongly suggests that your sample is not representative of the inferential population. How did you deal with partially completed responses?
The fact that you finished with a 3-factor solution which was different to the original 4-factor solution means that you can’t rely on the work done by the original authors to assess construct, convergent or face validity. It might be that the clinical experts in this area do not believe that this scale is fit for purpose. It would have been more compelling if you had developed the scale with a portion of the sample and tested it on the holdout.
The final percentage of variance explained by approximately 58% which is not that much, suggesting that the data was not a good fit for your analysis. Normally, scales are validated using maximum likelihood or principal axis factoring. Was cross-loading present? What were the communalities? Did you delete any items in the original scale?
Given that you have a new scale, why did you not complete the CFA as well? This should be done at a minimum would provide the reader with more confidence that your data are a good fit for your scale.
Author Response
Dear Reviewer,
We would like to thank you for taking the time to review our paper and provide us with beneficial comments.
We performed CFA for the main dataset as we only translated the original EDE-Q without necessary modifications. As the results of CFA did not support the four-factor model of the original EDE-Q, we performed further analysis after data split by using EFA for model convergent suggestion and CFA for confirming the results of EFA.
All the required results were illustrated in 6 tables and 1 figure (including communalities, estimates, ..etc), kindly refer to them. Please note that our model does not have cross-loading and some items were removed because of their correlation value less than 0.4 and these were reported and stated in the manuscript and cutoff points were supported by evidence.
In addition, some associations were reported between CFA and EFA, and socio-demographics such as province and BMI. Therefore, the method and result sections were majorly revised and the discussion section was modified accordingly.
We are aware of the nature of our data with high proportion of females, and we acknowledged that as a limitation. In the introduction section we clarified the importance of using EDE-Q as a gold standard tool for eating disorder screening and diagnosis. In the conclusion we recommended further study testing discrimination validity as our study focus only on convergent validity.
For more details please see the revised version manuscript.
Again, thank you for your valuable comments!
Regards,
Authors
Reviewer 2 Report
Dear Authors,
the present study examines the factor structure of the Arabic Version of the Eating 2
Disorder Examination Questionnaire (EDE-Q).
The paper is well-written, and the study is interesting as it may provide researchers and practitioners with a valid and reliable tool to be used for the assessment of eating disorders in the Arabic population.
However there are some shortcomings that concern me and limit its potential applications.
- Firstly, the title and the abstract. This study provides evidence on the factor structure of the EDE-Q in the Arabic population, while there is no evidence to endorse that this tool is a valid and reliable tool (e.g., discriminant and convergent validity).
- Moreover, the rules for conducting a translation/ validation study when there is a clear and recognised theoretical framework behind the tool require/suggest first conducting a Confirmatory Factor Analysis. This allows to verify whether the original structure of the tool can be confirmed.
Then, if your data do not support the original four-factor structure (the latter should be clear from the abstract), it can be conducted an exploratory factor analysis to explore the factor structure. When the sample is large enough, the sample can also be split into two, so conducting EFA on the first split-half sample and CFA on the second split-half sample. This procedure can give more robustness to your three-factor structure or even reveal that the original four-factor structure is confirmed from the beginning. Anyway, important information (e.g., communalities, explained variance) lacks from the EFA analysis.
- the introduction section lacks meaningful information on the other tools available for assessing eating disorders.
I can see a long list, but the authors should consider the need to explain why we need the EDE-Q in Arabic. Why specifically this tool, rather than other tools available in Arabic? Why is it a gold standard?
- the tool should be better described in general. For example, did the study allow to identify clinical cases (cut-off points?)
- why the authors have not explored the differences in EDE-Q subscales according to socio-demograpics?
- The limitation section should be strengthened as the discussion on the implications for research and interventions.
I hope the authors would like to work on my comments/suggestions to enhance the quality of their important work.
Sincerely
Author Response
Dear Reviewer,
We would like to thank you for taking the time to review our paper and provide us with beneficial comments. Kindly, note that we performed CFA for the main dataset as we only translated the original EDE-Q without necessary modifications. As the results of CFA did not support the four-factor model of the original EDE-Q, we performed further analysis after data split by using EFA for model convergent suggestion and CFA for confirming the results of EFA. All the required results were illustrated in 6 tables and 1 figure (including communalities, estimates, ..etc), kindly refer to them. Please note that our model does not have cross-loading and some items were removed because of their correlation value less than 0.4 and these were reported and stated in the manuscript and cutoff points were supported by evidence. In addition, some associations were reported between CFA and EFA, and socio-demographics such as province and BMI. Therefore, the method and result sections were majorly revised and the discussion section was modified accordingly. We are aware of the nature of our data with high proportion of females, and we acknowledged that as a limitation. In the introduction section we clarified the importance of using EDE-Q as a gold standard tool for eating disorder screening and diagnosis. In the conclusion we recommended further study testing discrimination validity as our study focus only on convergent validity.
For more details please see the revised version manuscript.
Again, thank you for your valuable comments!
Regards,
Authors
Reviewer 3 Report
Authors should clarify their results and clarify their methodological options, namely:
Homogeneity statistics of items and coefficients of internal consistency (α) from the Eating Disorder Examination Question-150 naire (EDE-Q 6.0), by item and by factor: Average SD r corrected; Cronbach's; α corrected Variation of each of the factors and the total of the Scale from the Eating Disorder Examination Question-150 naire (EDE-Q 6.0) with the independent variables (age, sex, nationality, province, etc) Pearson’s correlation between the factors and the total of the Adolescent Students’ Attitudes Scale towards Sexuality.
The discussion deserved a more detailed and in-depth analysis by items and factors extracted.
The conclusions could be improved, namely with the implications for clinical practice.
Author Response

(The authors gave the same response as above.)

Round 2
Reviewer 2 Report
The paper is significantly improved. Good luck with your work.
Sincerely!
Reviewer 3 Report
The authors carried out the recommendations indicated by me.